# LEARNING TO ACT WITH AFFORDANCE-AWARE MULTIMODAL NEURAL SLAM

## ABSTRACT

Recent years have witnessed an emerging paradigm shift toward embodied artificial intelligence, in which an agent must learn to solve challenging tasks by interacting with its environment. There are several challenges in solving embodied multimodal tasks, including long-horizon planning, vision-and-language grounding, and efficient exploration. We focus on a critical bottleneck, namely the performance of planning and navigation. To tackle this challenge, we propose a Neural SLAM approach that, for the first time, utilizes several modalities for exploration, predicts an affordance-aware semantic map, and plans over it at the same time. This significantly improves exploration efficiency, leads to robust long-horizon planning, and enables effective vision-and-language grounding. With the proposed Affordance-aware Multimodal Neural SLAM (AMSLAM) approach, we obtain more than $40\%$ improvement over prior published work on the ALFRED benchmark and set a new state-of-the-art generalization performance at a success rate of $23.48\%$ on the test unseen scenes.

## 1 INTRODUCTION

There is significant recent progress in learning simulated embodied agents Pashevich et al. (2021); Zhang & Chai (2021); Blukis et al. (2021); Nagarajan & Grauman (2020); Singh et al. (2020); Suglia et al. (2021) that follow human language instructions, process multi-sensory inputs and act to complete complex tasks Anderson et al. (2018); Das et al. (2018); Chen et al. (2019); Shridhar et al. (2020). Despite this, challenges remain before agent performance approaches satisfactory levels, including long-horizon planning and reasoning Blukis et al. (2021), effective language grounding in visually rich environments, efficient exploration Chen et al. (2018), and importantly, generalization to unseen environments. Most prior work Singh et al. (2020); Pashevich et al. (2021); Nguyen et al. (2021); Suglia et al. (2021) adopted end-to-end deep learning models that map visual and language inputs into action sequences. Besides being difficult to interpret, these models show limited generalization, suffering from significant performance drop when tested on new tasks and scenes.

In contrast, hierarchical approach Zhang & Chai (2021); Blukis et al. (2021) achieve better generalization performance and interpretability. Although hierarchical structure is helpful for long-horizon planning, its key impact is an expressive semantic representation of the environment acquired via Neural SLAM-based approaches Chaplot et al. (2020a;c); Blukis et al. (2021). However, a missing component in these methods is fine-grained affordance Kim & Sukhatme (2015); Qi et al. (2019). To build a robotic assistant that can follow human instructions to complete a task (e.g., *Open the fridge and grab me a soda*), it is essential that the agent can perform affordance-aware navigation: it must navigate to a reasonable position and pose near the fridge that enables follow-on actions *open* and *pick-up*. Operationally, the agent has to move to a location where the fridge is within reach yet without arresting the fridge door from being opened. Ideally, it should also position itself so that the soda is in its first person viewing field to allow the follow-on *pick-up* action. This is challenging compared to pure navigation (where navigating to any location close to the fridge is acceptable). To achieve this, **we propose a sophisticated affordance-aware semantic representation that leads to accurate planning for navigation setting up subsequent object interactions for success.**

Efficient exploration of the environment Ramakrishnan et al. (2021); Chen et al. (2018) needs to be addressed to establish this semantic representation - it is unacceptable for a robot to wander around for an extended period of time to complete a single task in a real-world setting. To resolve

this issue, **we propose the first multimodal exploration module that takes language instruction as guidance and keeps track of visited regions to explore the area of interest effectively and efficiently.** This lays a foundation for map construction, which is critical to long-horizon planning.

Here, we introduce Affordance-aware Multimodal Neural SLAM (AMSLAM), which implements two key insights to address the challenges of robust long-horizon planning, namely, efficient exploration and generalization: **1. Affordance-aware semantic representation** that estimates object information in terms of where the agent can interact with them to support sophisticated affordance-aware navigation, and **2. Task-driven** multimodal exploration that takes guidance from language instruction, visual input, and previously explored regions to improve the effectiveness and efficiency of exploration. AMSLAM is the first Neural SLAM-based approach for Embodied AI tasks to utilize several modalities for effective exploration and an affordance-aware semantic representation for robust long-horizon planning. We conduct comprehensive empirical studies on the ALFRED benchmark Shridhar et al. (2020) to demonstrate the key components of AMSLAM, setting a new state-of-the-art generalization performance at $23.48\%$, a $>40\%$ improvement over prior published state-of-the-art approaches.

## 2    RELATED WORK

Recent progress in Embodied Artificial Intelligence, spans both simulation environments Kolve et al. (2017); Li et al. (2021); Savva et al. (2019); Gan et al. (2020); Puig et al. (2018) and sophisticated tasks Das et al. (2018); Anderson et al. (2018); Shridhar et al. (2020). Our work is most closely related to research in language-guided task completion, Neural SLAM, and exploration.

**Language-Guided Task Completion**. ALFRED Shridhar et al. (2020) is a benchmark that enables a learning agent to follow natural language descriptions to complete complex household tasks. The agent's goal is to learn a mapping from natural language instructions to a sequence of actions for task completion in a simulated 3D environment. Various modeling approaches have been proposed falling into roughly two families of methods. The first focuses on learning large end-to-end models that directly translate instructions to low-level agent actions Singh et al. (2020); Suglia et al. (2021); Pashevich et al. (2021). However, these agents typically suffer from poor generalization performance, and are difficult to interpret. Recently, hierarchical approaches Zhang & Chai (2021); Blukis et al. (2021) have attracted attention due to their better generalization and interpretability. We also adopt a hierarchical structure, focusing on affordance-aware navigation thereby achieving significantly better generalization than all existing approaches.

**Neural SLAM and Affordance-aware Semantic Representation**. Neural SLAM Chaplot et al. (2020a;b;c), constructs an environment semantic representation enabling map-based long-horizon planning Chaplot et al. (2021). However, these are tested in pure navigation tasks instead of complex household tasks, and does not consider affordance Qi et al. (2019); Nagarajan & Grauman (2020); Xu et al. (2020), which is required for tasks involving both navigation and manipulations. In Blukis et al. (2021), the authors utilize SLAM for 3D environment reconstruction in language-guided task completion. Their approach relies heavily on accurate depth prediction (less robust in unseen environments). Instead, we propose a waypoint-oriented representation which associates each object with the locations on the floor from where the agent can interact with the object. Furthermore, different from the 2D affordance map in Blukis et al. (2021) that directly predicts affordance type, our semantic representation supports more fine-grained control of the robot's position and pose, which facilitates significantly better generalization. The approach in Qi et al. (2019) assumes direct access to the ground truth depth information (not available in our setup) and the method in Nagarajan & Grauman (2020) only focuses on pure navigation problems.

**Learning to Explore for Navigation**. An essential step in Neural SLAM-based approaches is learning to explore the environment for map building Ramakrishnan et al. (2021); Chen et al. (2018); Jayaraman & Grauman (2018); Chaplot et al. (2020a). Multiple approaches have been proposed to tackle aspects of exploration in the reinforcement learning Schmidhuber (1991); Pathak et al. (2017); Burda et al. (2018); Chen et al. (2018); Jayaraman & Grauman (2018), computer vision Ramakrishnan et al. (2021); Nagarajan & Grauman (2020), and robotics Blukis et al. (2021); Harrison et al. (2018) communities. The central principle of prior methods is learning to reduce environment uncertainty; different definitions of uncertainty lead to the following types of methods Ramakrishnan et al. (2021). Curiosity-driven Schmidhuber (1991); Pathak et al. (2017); Burda et al. (2018) approaches learn forward dynamics and reward visiting areas that are poorly predicted by the model. Count-based exploration Tang et al. (2017); Bellemare et al. (2016); Ostrovski et al. (2017); Rashid

et al. (2020) encourages visiting states that are less frequently visited. Coverage-based Chen et al. (2018); Jayaraman & Grauman (2018) approaches reward visiting all navigable areas by searching in a task-agnostic manner. In contrast, we propose a multimodal exploration approach utilizing egocentric visual input, language instructions, and memory of explored areas to reduce task-specific uncertainty of points of interest (areas important to complete the task). We show this to be more efficient, leading to more effective map prediction and robust planning.

## 3 PROBLEM FORMULATION

We focus on the ALFRED challenge Shridhar et al. (2020), where an agent is asked to follow human instructions to complete long-horizon household tasks in indoor scenes (simulated in AI2Thor Kolve et al. (2017)). Each task in ALFRED consists of several subgoals for either navigation (moving in the environment) or object interactions (interacting with at least one object). Language inputs contain a high-level task description and a sequence of low-level step-by-step instructions (each corresponding to a subgoal). The agent is a simulated robot with access to the states of the environment only through a front-view RGB camera with a relatively small field of view. The agent's own state is a 5-tuple $(x, y, r, h, o)$, where $x, y$ are its 2D position, $r$ the horizontal rotation angle, $h$ the vertical camera angles (also called "horizon") and $o$ the type of object held in its hand. The state space of the agent is discrete, with navigation actions: MoveAhead (moving forward by $0.25m$), RotateLeft & RotateRight (rotating in the horizontal plane by $90°$) and LookUp & LookDown (adjusting the horizon by $15°$). Formally, $r \in \{0°, 90°, 180°, 270°\}$ and $h \in \{60°, 45°, ..., -15°, -30°, \}$ where positive $h$ indicates facing downward. With these discrete actions, the agent has full knowledge of the relative changes $\Delta x, \Delta y, \Delta r$ and $\Delta h$. Each of the 7 object interaction actions (PickUp, Open, Slice, etc.) is parametrized by an binary mask for the target object, which is usually predicted with a pre-trained instance segmentation module. Featuring long-horizon tasks with a range of interactions, the ALFRED challenge evaluates an agent's ability to perform tasks over *unseen test scenes*, while only allowing $\leq 1000$ steps and $\leq 10$ action failures for each task at inference time.

## 4 AFFORDANCE-AWARE MULTIMODAL NEURAL SLAM

Affordance-aware navigation is a major challenge in solving complex and long-horizon indoor tasks such as ALFRED with both navigation and object interactions. Specifically, given each object of interest in the scene, the agent is required to not only find and approach it but also end up at a pose $(x, y, r, h)$, that is feasible for subsequent interactions with the object. For instance, to open a fridge, the robot should approach the fridge closely enough (so the door is within reach), look at it (so that the fridge is in the field of view), and leave enough room to open the door. To solve a long-horizon task involving multiple navigation and object interaction subgoals, it is natural to use an explicit semantic map, either 2D or 3D, of the environment (similar to Neural Active SLAM Chaplot et al. (2020a)), together with model-based planning (e.g. as in HLSM Blukis et al. (2021)). This line of work tends to generalize better than models that directly learn mappings from human instructions to navigation & interaction actions (e.g., E.T. Pashevich et al. (2021)). With perfect knowledge of the environment, it is possible to achieve (nearly) perfect performance. In practice, however, the semantic map acquired at inference time is usually far from ideal, primarily due to **Incompleteness** (missing information due to insufficient exploration of the scene) and **Inaccuracy** (erroneous object location prediction on the map, especially for small objects).

To improve exploration performance, we propose a multimodal module that, at each step, predicts an exploration action $a \in \{\text{MoveAhead}, \text{RotateLeft}, \text{RotateRight}\}$ by taking visual observations & actions in the past, step-by-step language instructions, and the explored area map which indicates where the agent has visited. We show that, compared to existing model-based approaches on ALFRED (e.g., HLSM Blukis et al. (2021) which applies random exploration), our use of low-level language instructions leads to more efficient exploration. The proposed exploration module operates at the subgoal level and only predicts exploration actions (in contrast to E.T. which directly predicts actions for the entire task). The extra modality (the explored area) facilitates exploration by providing the agent with explicit spatial information. We illustrate the exploration module in Figure 3, elaborate its details in Section 4.3, and empirically demonstrate its advantages in Section 5.

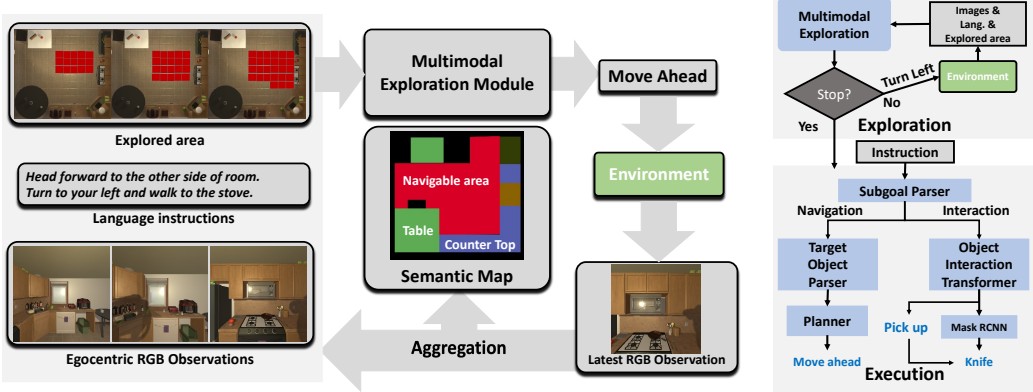

Exploration Flow                    Execution Flow

Figure 1: **Exploration Flow**: The agent aims to explore the environment given the guidance from low-level language instructions, previous exploration actions, and the explored area where the agent has visited and made observations. **Execution Flow**: Given the language instructions and the affordance-aware semantic representation (i.e., semantic map) acquired during exploration, the agent executes the subgoals sequentially. It uses a planning module (which consumes the semantic map) for navigation subgoals and an object interaction transformer for other subgoals.

To deal with the inaccuracy in map prediction, we carefully design an affordance-aware semantic representation for the environments. On one hand, knowing the precise spatial coordinates of objects requires precise depth information, which is difficult to acquire due to 3D sensor noise and/or inaccuracy in predicting depth from 2D images. On the other hand, affordance-aware navigation essentially asks for poses $(x, y, r, h)$ of the agent suitable for interactions with the target objects, thus requiring only coarse-grained spatial information. Given an object type $o$, we define such corresponding poses as *waypoints* $\mathcal{W}_o$ and then treat navigation as a path planning problem among different waypoints. To generate such waypoints, we handle large objects (fridges, cabinets, etc.) and small objects (apples, mug, etc.) differently. The waypoints for large objects are computed using 2D grid maps predicted and aggregated from front-view camera images by a CNN-based network; for small objects, we directly search over all observations acquired during the exploration phase with the help of a pre-trained Mask RCNN He et al. (2017) (detailed below in Section 4.2).

## 4.1 Overall Pipeline

We illustrate the overall inference pipeline of our proposed framework in Figure 1. Given a task $\mathcal{T}$ specified by a high-level goal description and low-level human instructions, our method proceeds in two phases: exploration and execution. During exploration, the agent navigates across the room (guided by the language instructions) for a sufficient exploration of the indoor scene, where a multimodal transformer predicts the exploration actions sequentially. In the meantime, an affordance-aware semantic representation is acquired given the egocentric observations (images) at each step by a carefully designed neural SLAM system. This representation can be used to derive *waypoints* for all target objects (from which the agent can interact with the objects of interest). When the exploration phase ends (by the agent predicting Stop), it moves to the execution phase, where the agent carries out actions predicted for each subgoal of the task $\mathcal{T}$ in a sequential manner. Specifically, since each step-by-step language instruction corresponds to a subgoal, we use a Transformer-based Vaswani et al. (2017) subgoal parser to process the text and predict the subgoal $g \in \mathcal{T}$ is for navigation or not. For a navigation subgoal, we further use another Transformer-based target object parser to process the same text and predict the categories of the target objects, which are consumed together with the affordance-aware semantic representation by a Dijkstra-based planner to generate navigation actions. Otherwise, an object interaction transformer takes charge of the action predictions given the visual and language inputs of the object interaction subgoal.

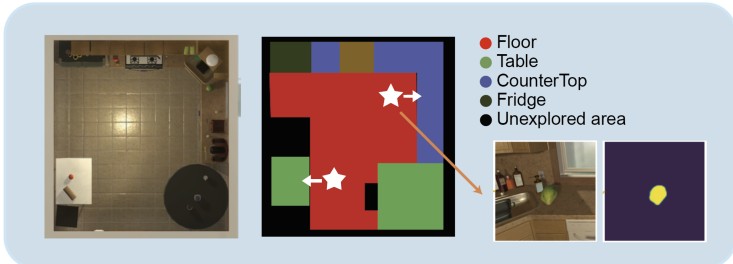

Figure 2: **Illustration of the affordance-aware semantic representation used in our framework.** **(left)** A top-down view of an indoor scene used in ALFRED (this view is not available to the agent at test time). **(middle)** A visualization of the corresponding semantic map; only shown are the side table (in green), counter top (in blue) and the navigable area (in red). Two waypoints (drawn in white stars and arrows) are displayed for side table ($\in$ large object) and lettuce ($\in$ small object), respectively. **(right)** While the waypoint for side table is computed from the predicted map, the waypoint for lettuce is obtained by searching among all exploration steps. The visual observation and its mask prediction on the waypoint for lettuce are shown. For reference, the high-level goal description of the task used in this example is "pick up the lettuce and place it on a table".

## 4.2 AFFORDANCE-AWARE SEMANTIC REPRESENTATION

We empirically show in Section 5.1 that a major bottleneck for solving long-horizon navigation & interaction tasks is the affordance-aware navigation. To do so, the agent needs a position and pose (defined as *waypoints* previously in this section) from which the potential follow-on actions for the target object are feasible, rather than the exact location of the target object. Accordingly, our goal is to develop a map representation that supports waypoint generation so that navigation can be solved reasonably well by path planning. ALFRED supports more than 100 types of objects (one way in which it mimics real-world complexity), and we propose to handle small and large objects differently. We detail our design below and give an example in Figure 2. Further implementation details are available in the Appendix.

For a class $c_{large}$ out of $N_{large}$ large object types, we compute its waypoint $(x^*, y^*, r^*, h^*)$ in 3 steps. **First**, we find all positions $\{(x, y)\}$ that might contain an instance of class $c_{large}$ and all navigable locations for the robot, both represented on a 2D grid map of dimension $G \times G \times (N_{large} + 1)$, with grid size $G = 37$ and unit length $0.25m$. Each grid point in the map has a binary multi-hot vector to represent whether each object class appears there and whether the point is navigable. Specifically, at each $(x, y, r)$ visited in the exploration phase, we use a pre-trained CNN (whose inputs are images at 3 different horizons observed at $(x, y, r)$) to predict a small partial map of the 2D map. We then aggregate across all exploration steps by max-pooling over these partial maps, each translated and rotated via a Spatial Transformer Jaderberg et al. (2015). Similar to Neural Active SLAM Chaplot et al. (2020a), as we know the changes $(\Delta x, \Delta y, \Delta r)$ of the agent after each action, we can directly compute the parameters of the Spatial Transformer. After applying some post-processing, we obtain the final 2D map estimation. Details about the CNN architecture, its pre-training process and the post-processing steps are in the Appendix. **Second**, we find $(p_x, p_y)$ on this 2D map as the most confident position predicted for class $c$ and then find the navigable position $(x', y')$ closest to $(p_x, p_y)$. **Third**, we choose the rotation $r^*$ to be the one, suppose the agent stands at $(x', y')$, that an object at $(p_x, p_y)$ appears closest to the center of the agent's field of view. To leave room for object interactions (by the agent's backing up a few steps), we compute $x^* = x' - \delta_1(c, r^*)$ and $y^* = y' - \delta_2(c, r^*)$ where $\delta_*$ are rule-based functions to (see Appendix A.2). We defer the estimation of the horizon $h^*$ (the camera angle relative to the horizontal) to the execution phase described in Section 4.4.

For a small object type $c_{small}$, predicting its 3D coordinates precisely is rather challenging (even 2D object detection for small objects is hard Kisantal et al. (2019)). In ALFRED, this is especially true since only RGB images are given at test time and many types of small objects occur rarely. To deal with this challenge, we propose to directly find the waypoint $(x^*, y^*, r^*, h^*)$ for $c_{small}$ by searching through all observations (RGB images) at each step during exploration. Specifically, we compare all instance masks for $c_{small}$ predicted by a pre-trained Mask RCNN He et al. (2017)

that are of confidence $\geq \tau_c$. Then $(x^*, y^*, r^*, h)$ is computed as the one where the aforementioned mask prediction has the largest area. Similar to the waypoint generation for large objects, we do not estimate the horizon $h^*$ until the execution phase. Directly finding the waypoint of $c_{small}$ (without estimating its location) relies heavily on how well the exploration is carried out (discussed later in Section 4.3). In case the agent finds no such waypoint (no valid observation of $c_{small}$ during exploration), we instead use the waypoint for the container (normally of large object type such as fridge, side table) of the small object, which is much easier to find. Most small objects appear on (or are contained in) containers, whose object types are predicted by the target object parser (see Section 4.4). While the proposed affordance-aware semantic representation is critical for robust long-horizon planning, we also add a brief discussion on its limitations in the Appendix.

## 4.3 TASK-DRIVEN MULTIMODAL EXPLORATION

The multimodal exploration module consists of several sub-modules, either learned or pre-trained/fixed. At a high level, given a task, the exploration go through its navigation subgoals one by one, with the subgoal parser (see Section 4.4) predicting whether a subgoal is for navigation. Specifically, the module predicts exploration actions $a \in \{\text{MoveAhead}, \text{RotateLeft}, \text{RotateRight}\}$ or Stop auto-regressively for each navigation subgoal. The agent switches to the next navigation subgoal whenever it predicts Stop until the last one to end the exploration phase. Instead of random exploration (as in HLSM) or exploration for maximum coverage (as in Active Neural SLAM), our module utilizes low-level language instructions to achieve task-driven exploration for better efficiency. In ALFRED the exploration is done individually for each task and the steps count towards the total steps (which has a limit of 1000).

There are 4 modalities (and 4 corresponding branches). At each time step, the first branch takes two low-level human instructions, one for the current navigation subgoal, and one for the subsequent object interaction subgoal. The second and third branches consume the egocentric observations (images) and the previous exploration actions, respectively. Inspired by E.T. Pashevich et al. (2021), we use a cross-modal transformer (also see Lu et al. (2019); Zhang & Chai (2021)), which aggregates inputs of these 3 modalities across all previous exploration steps to output $f_{1,2,3}$. The fourth modality, the explored area, is a $G \times G$ 2D grid map marked 1 for the regions observed by the agent in the past and elsewhere 0, except that the center of the map (which always indicates the agent's position) is marked 2. This map is constructed in 2 steps. First, at each previous exploration step, since the agent can only observe a small area in front of it, we define a single-step explored region as a binary map where there is a $5 \times 3$ rectangle grid of 1's representing where the agent has observed (see details in Appendix). Second, we aggregate the maps across all previous steps. Each single-step map is translated and rotated by a Spatial Transformer and then merged by max-pooling (similar to the semantic map described in Section 4.2). Given this explored area map, we use a CNN to extract $f_4 \in \mathbb{R}^{32}$. Together with $f_{1,2,3}$, we finally use a MLP to predict the next exploration action. Illustrated in Figure 4.3 and evaluated in Section 5.3, our design to represent the action history explicitly and geometrically improves both the effectiveness and efficiency of the exploration.

We train the multimodal exploration module supervisedly. A common practice (e.g. in HLSM Blukis et al. (2021), EmBERT Suglia et al. (2021), LWIT Nguyen et al. (2021)) during inference is to augment the exploration by injecting actions periodically. We manually inject 4 RotateRight after every 2 MoveAhead predicted by our module to acquire 360° views of the scenes. Moreover, the semantic representation produced by the neural SLAM (in Section 4.2) requires input images of 3 different horizons. So we further inject two LookUp or two LookDown actions alternately after every exploration action. This zigzagging scheme is an efficient way to acquire images of multiple horizons, which only triples the total number of steps in the exploration phase. Since each exploration step counts towards the total steps for a task, there is a trade-off between exploration (a better view of the environment) and exploitation (there is an upper limit of 1000 for total steps). Full implementation details and examples are provided in the Appendix.

## 4.4 OTHER MODULES

**Subgoal Parser & Target Object Parser** Both the subgoal parser and the target object parser take the language instruction as the only input. Both models use the same Transformer-based architecture (not sharing weights, though) and are trained supervisedly. The subgoal parser performs a binary

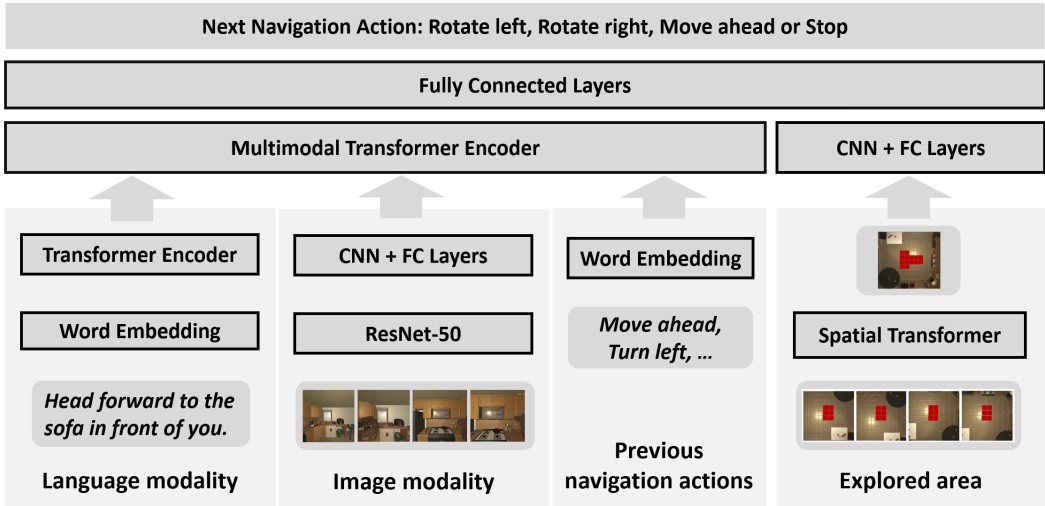

Figure 3: **The exploration module.** Consisting of several transformer-based networks, the exploration module operates at a subgoal level. Given a navigation subgoal during inference, it takes multiple modalities as input including language instructions for the current and subsequent subgoals, egocentric observations from the agent, and exploration actions in the past as well as their corresponding explored area. It predicts the next exploration action carried out in the environment auto-regressively. Illustrated is an example where given 4 previous actions the model tries to predict the 5-th. Note that for brevity the positional (and temporal) encoding layers right before the transformer encoders are omitted in the figure.

classification, predicting whether each subgoal is for navigation (based on its human instruction). The target object parser predicts both the target object and its container (if it has one) by taking the language instructions for the navigation subgoal *and for the next subgoal*.

**Online Planner for Affordance-aware Navigation** Given the affordance-aware semantic representation that supports waypoints generation, we deal with a navigation subgoal in 3 steps: (1) Obtain a waypoint $(x, y, r, *)$ for the object type predicted by the target object parser. (2) Derive an action sequence from the path connecting the current location and pose $(x', y', r', h')$ to $(x, y, r, h')$ using Dijkstra's algorithm. (3) Decide the horizon $h$ by online exploration by first navigating to $(x, y, r, h')$. The agent then goes through 6 horizons $\{60°, 45°, ..., 0°, -15°\}$ (essentially a search over most horizons allowed for the agent) and obtains the mask prediction with confidence $> 0.8$ (selected from $\{0.5, 0.6, 0.7, 0.8, 0.9\}$ using the valid unseen data) of the target object type by a pre-trained Mask RCNN. Finally, we select $h$ with the largest mask area. See ablation studies of this scheme in Section 5.3. We also include a two-step backtracking mechanism for our planner (with more details in the Appendix).

**Object Interaction Transformer** We adopt the cross-modal transformer again at the subgoal level for object interaction subgoals, which maps language instructions and visual observations (only for the current object interaction subgoal) to actions. The module is trained by imitation learning on the ALFRED training data at the subgoal level.

## 5 EXPERIMENTS

We evaluate our method for language-guided task completion on the ALFRED challenge Shridhar et al. (2020), which supports task evaluation in unseen environments (i.e., room layouts), the main focus of our method. ALFRED provides both a validation (with ground truth provided) and a test set (evaluation occurs in the server) for tasks sampled in indoor scenes unseen during training. The training split contains 21,023 tasks sampled from 108 scenes (i.e., rooms), the valid unseen split and the test unseen split contain 821 tasks sampled from 4 scenes and 1,529 tasks sampled from 8 scenes, respectively. For all experiments, we report the commonly used evaluation metrics: the task

level *Success Rate* and the subgoal level *Goal Condition*. Notice that each task in ALFRED allows up to 1000 total steps (including both exploration and execution phases in our method).

We organize the presentation of the numerical results into 3 parts. Firstly, we demonstrate that the affordance-aware navigation is the major bottleneck for language-guided task completion. Secondly, we present our main numerical results showing that with better affordance-aware navigation, our method significantly outperforms previous state-of-the-art methods on ALFRED. Finally, we perform a series of ablation studies to justify our design.[1]

## 5.1 Affordance-aware Navigation

We analyze the generalization performance of our approach by using ground truth actions for navigation subgoals and actions generated by the object interaction transformer otherwise. Similar to solving indoor tasks in real-world settings, the major bottleneck in tackling ALFRED is affordance-aware navigation. A navigation subgoal succeeds only if the agent stays close enough to the target object (so that it is within reach), sets a camera angle so that the object is in the field of view, and leaves room for object articulation (e.g., open a door of a fridge). We perform several experiments on the valid unseen data of ALFRED with numerical results reported in Table 1. Specifically, when the ground truth navigation actions (denoted GT navi.) are used during inference, our framework achieves an extremely high success rate (64.6). However, the performance drops significantly if perturbations (random displacement is adding $\pm 1$ to the coordinates) are added to the target $(x, y, r, h)$ of each navigation subgoal, verifying our claim about the need for affordance-aware navigation.

|  | Valid Unseen (%) | |
| --- | --- | --- |
|  | Success Rate | Goal Cond. |
| HLSM Blukis et al. (2021) | 11.8 | 24.7 |
| GT navi. + Obj. Int. Transformer | 64.6 | 74.2 |
| GT navi. + Obj. Int. Transformer + rand. horizon | 21.7 | 35.9 |
| GT navi. + Obj. Int. Transformer + rand. displacement | 47.3 | 65.4 |

Table 1: **Generalization performance of our framework with ground truth navigation (denoted GT navi.) inserted together with different types of pose perturbations.** For reference, we also include results from HLSM Blukis et al. (2021), the previous state-of-the-art method on ALFRED.

## 5.2 Main Results

We present the main results of our proposed method, evaluated on the test unseen and valid unseen data of ALFRED. Our framework uses no ground truth or metadata about the environment during inference. As introduced in Section 4.1, our framework performs exploration to acquire knowledge about the indoor scenes and then predict actions in a hierarchical approach with both a rule-based planner and a few learning-based modules. We set a new state-of-the-art performance with a substantial improvement ($>40\%$) over previously published methods (see Table 2). See additional qualitative results in Appendix.

|  | Test Unseen (%) | | Valid Unseen (%) | |
| --- | --- | --- | --- | --- |
|  | Success rate | Goal Cond. | Success rate | Goal Cond. |
| EmBERT Suglia et al. (2021) | 7.52 | 16.33 | 5.73 | 15.91 |
| E.T.+ Pashevich et al. (2021) | 8.57 | 18.56 | 7.32 | 20.87 |
| LWIT Nguyen et al. (2021) | 9.42 | 20.91 | 9.70 | 23.10 |
| HiTUT Zhang & Chai (2021) | 13.87 | 20.31 | 12.44 | 23.71 |
| ABP Kim et al. (2021) | 15.43 | 24.76 | - | - |
| HLSM Blukis et al. (2021) | 16.29 | 27.24 | 11.80 | 24.70 |
| AMSLAM (ours) | **23.48** | **34.64** | **17.68** | **33.96** |

Table 2: **Performance on valid & test unseen data from ALFRED.** Our method achieves new state-of-the-art results with a substantial improvement over previously published methods.

---

[1]The code to reproduce all results in this work will be made publicly available.

While AI2THOR adopts a discrete action/state space for the agent, we believe the idea behind our multimodal exploration design and waypoint-based semantic representation is applicable to generic embodied tasks. In the presence of motion noise and pose sensor noise, Active Neural SLAM shows that such inaccuracy can be solved to an acceptable level by simply learning a pose estimator.

## 5.3 ABLATION STUDIES

We perform ablation studies to justify our framework design. In Ablation Studies I, we examine the components in our multimodal exploration module. We choose four variants. In variant AMSLAM + rand. exploration, we replace the multimodal exploration by a random exploration strategy similar to HLSM. In variant AMSLAM - lang., we do not use the language instructions at all to guide the exploration process In variant AMSLAM - lang. (partial), we only use language instructions associated with the navigation subgoal (i.e., without the next object interaction subgoal) to guide the exploration. In variant AMSLAM - explored area, we do not use the extra modality in the exploration module. For all variants, the exploration phase ends when a pre-defined upper limit of action fails or exploration steps is reached. In Table 3, we compare the valid unseen performance on ALFRED, the coverage, defined as the number of distinct $(x, y)$ pairs visited during the exploration phase (average per task), and the coverage efficiency (Cov. Eff.), defined as the coverage divided by total number of (un-augmented) exploration steps. More details are available in the Appendix.

In Ablation Studies II, we examine the components in the affordance-aware map representation and in the online planner. We justify our approach of handling small (denoted as instance mask waypoints) and large objects (denoted as object map waypoints) in different ways by showing that the generalization performance degrades drastically when adopting only one strategy for waypoint generation. We show that the backing up rule in the planner (to leave room for articulated objects) is important. Moreover, we evaluate a variant (AMSLAM + rand. horizon) where horizon $h$ is perturbed for all subgoals to justify our online exploration (and backtracking) strategy for finding the best $h$. See numerical results in Table 4 and implementation details in Appendix.

| | Valid Unseen (%) | | Coverage Analysis | |
|---|---|---|---|---|
| | Success Rate | Goal Cond. | Coverage | Cov. Eff. (%) |
| AMSLAM + rand. exploration | 9.73 | 22.10 | 20.40 | 58.39 |
| AMSLAM - lang. | 4.30 | 15.90 | 9.10 | 43.50 |
| AMSLAM - lang. (partial) | 13.66 | 28.93 | 24.37 | 66.05 |
| AMSLAM - explored area | 15.17 | 30.90 | 27.13 | 65.48 |
| AMSLAM (ours) | **17.68** | **33.96** | **28.70** | **67.09** |

Table 3: Ablation Studies I: the multimodal exploration module in AMSLAM.

| | Valid Unseen (%) | |
|---|---|---|
| | Success Rate | Goal Cond. |
| AMSLAM - object map waypoints | 12.10 | 26.73 |
| AMSLAM - instance mask waypoints | 8.90 | 18.70 |
| AMSLAM - back up steps | 12.50 | 28.85 |
| AMSLAM + rand. horizon | 9.81 | 20.60 |
| AMSLAM (ours) | **17.68** | **33.96** |

Table 4: Ablation Studies II: affordance-aware semantic representation & planner in AMSLAM.

## 6 CONCLUSION

This work presents comprehensive empirical results that substantiate the importance of affordance-aware navigation for language-guided task completion. We propose Affordance-aware Multimodal Neural SLAM (AMSLAM) that constructs an accurate affordance-aware semantic representation and collects data efficiently through a novel multimodal exploration module. We conduct thorough ablation studies to demonstrate that the various aspects of our design choices is essential to the performance. Last but not least, AMSLAM achieves more than $40\%$ improvement over prior published work on the ALFRED benchmark and sets a new state-of-the-art generalization performance at a success rate of $23.48\%$ on test unseen scenes.

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

## A  VALIDATION OF OBJECT INTERACTION TRANSFORMER

To begin with, we first validate the hierarchical approach (i.e., subgoal level task execution) adopted by our method. We takes a hierarchical approach where each subgoal is executed sequentially by either a Dijkstra-based planner (for navigation subgoals) or the object interaction transformer (otherwise). In this section, we justify the use of the object interaction transformer, which is a cross-modal transformer trained at the subgoal level. In specific, we pre-process the training data from the original training fold of ALFRED such that each trajectory contains inputs (low-level language instructions and visual observations) and ground truth actions for a single object interaction subgoal. We evaluate our proposed module on the valid unseen split of ALFRED. Compared to E.T.+, which adopts a cross-modal transformer trained in full task level and with extra synthesized data, the object interaction transformer generalize better on nearly all subgoals, as reported in Table 5.

| | Valid Unseen (%) | | | | | | |
|---|---|---|---|---|---|---|---|
| | Toggle | Pickup | Cool | Put | Heat | Clean | Slice |
| E.T.+ Pashevich et al. (2021) | 83.2 | 69.0 | 99.1 | 69.6 | **99.3** | 91.2 | 65.8 |
| Obj. Int. Transformer (ours) | **86.1** | **72.0** | **100.0** | **75.3** | 98.5 | **91.7** | **72.1** |

Table 5: **Success rates (%) of all 7 object interaction subgoals evaluated on valid unseen data in ALFRED.** Overall, our method (object interaction transformer) generalizes better than E.T.+.

## B  AFFORDANCE-AWARE SEMANTIC REPRESENTATION

### B.1  NEURAL SLAM MODULE

**CNN Architecture**  The CNN used for predicting the 2D grid map of size $G \times G \times (N_{large} + 1)$ (with grid size $G = 37$ and unit length $0.25m$) has the following architecture. The inputs are 3 images, which are first processed by the ResNet-50 Faster RCNN feature extractor (we use the checkpoint provided by E.T.) Then the three 512-d features are each processed by the same 4 layer CNN (filter size 3, stride size 2, number of features as $256, 256, 256, 256$). The flattened and concatenated features from the previous step are then fed into a 4-layer FC network (number of features are $512, 512, 512, 128 \cdot 7 \cdot 10$). Next the output features are reshaped into a $128 \times 7 \times 10$ tensor. Finally another 3-layer CNN (filter size 3, stride size 1, number of features as $256, 256, N_{large} + 1$) takes in the tensor and output the $7 \times 10 \times (N_{large} + 1)$ map, which represents the prediction of a small region in front of the agent (in its egocentric view).

**Aggregation**   The 2D grid map predictions at each exploration step are aggregated via the use of a spatial transformer. The parameters to the spatial transformer are computed in the same way as the Neural Action SLAM (since we know exactly what each exploration action is and thus where the agent was headed). Each predicted 2D grip map is first rotated and translated by the spatial transformer (with the initial location considered as the center of the 2D map and the initial rotation angle considered as the direction facing upward) and then max-pooled to form the complete semantic map of the environment. During this process, since each region at a single step has a limited field of view, we mask the prediction at each step by a hard-coded $7 \times 10$ binary map (starting from the row the agent is standing at towards another 9 rows facing forward). We have tried multiple combinations for the shape of the binary map, with height $\in \{5, 6, 7\}$ and width $\in \{9, 10, 11\}$. We choose $7 \times 10$ using valid unseen data in ALFRED by the average prediction accuracy.

**Model Training**   We train the model by minimizing the cross-entropy distance between the ground truth semantic map and the predicted one. The training is performed for each single step map prediction. The ground truth for the navigable area is generated by using the API from AI2Thor. The ground truth for the object map is not available for scenes in ALFRED, which uses a version of AI2Thor that does not support bounding box information for general objects in the scenes. However, later AI2Thor versions support such functionality. There are some scene layout mismatches between later versions of AI2Thor and the version used by ALFRED, though. We solve this by manually inspecting all the 108 training scenes and fixing bugs by hand. We will release the code as well as the processed training data for our Neural SLAM module. We use the Adam optimizer to train the CNN model with an initial learning rate of $0.005$ and a linear decaying schedule (starting from the second half of the training) to 0 for a total of 10 epochs. We find the best checkpoint using the prediction accuracy evaluated on the valid seen and valid unseen data of ALFRED.

**Post-processing**   To have a more robust navigable area for long-horizon planning, we further apply some post-processing steps to the aggregated navigable map by the Neural SLAM module (i.e., the last dimension of the 2D grid map). Specifically, we consider map A as the binary navigable area map where only the predicted confidence greater than $0.95$ will be considered as a valid prediction for a navigable point (i.e., a value of 1). We also consider map B as the binary map where a location with greater or equal to 3 nearby points (i.e., the one whose L1 distance to it is 1) being navigable (here we use confidence threshold $0.5$) is marked as a navigable point. We then perform an element-wise product of the two maps to obtain the final navigable area.

### B.2   WAYPOINT GENERATION

**Small and Large Objects**   As mentioned in the paper, we handle small and large objects differently when designing our affordance-aware semantic representation. In specific, we consider large object types in the following list and otherwise small object types:

- armchair, chair, cart, sofa, shelf, drawer, cabinet, countertop, sink, stove burner
- fridge, bed, dresser, toilet, bathtub, ottoman, diningtable, sidetable, coffeetable, desk

**The Back up Steps**   The benefit of handling the large objects in a 2 step process is that we can compute their waypoints in a more flexible manner. In our framework, we find backing up a few steps to be a very effective strategy, which leaves some margin between the target position of the agent and the target object the agent needs to interact with This strategy is particularly critical for articulated objects. We use simple heuristics, denoted $\delta_1$ and $\delta_2$ as introduced in the paper. Specifically, the agent will back up 3 steps if the target object is a fridge, 2 steps if it is a safe, a cabinet or a drawer, and 1 step for everything else. The implementation of the $\delta_*(\cdot, \cdot)$ is simply an integer $1, 2$ or 3 for either $x$ or $y$ coordinate given the 4 different rotation angles $r \in \{0°, 90°, 180°, 270°\}$.

### B.3   LIMITATIONS

The existing framework regarding the affordance-aware semantic representation can be further improved by:

- Introducing an instance-level representation such that multiple instances of the same object type can be handled better.

Figure 4: An illustration of the single-step explored area for four exploration steps (**left**) and the aggregated one (**right**). The actual size of a single-step explored area is $5 \times 3$ instead of $3 \times 2$ shown here.

- Training with more room layouts to prevent overfitting as currently there are only 108 scenes in the training data of ALFRED.

## C   MULTIMODAL EXPLORATION

### C.1   MODEL TRAINING

**The Extra Modality: Explored Area**   The extra modality introduced in our multimodal exploration module essentially tracks the action history explicitly and geometrically. Specifically, during each step in the exploration phase, we hard-code a $5 \times 3$ binary mask to indicate where the agent has observed in the current egocentric view. We find the exact shape of this region does not matter much (we have tried 3x2, 4x2, 5x4) as long as it helps to track where the agent has visited. As the agent always stands at the center of the explored region map, we set the binary mask as starting from the center row and extending facing forward towards another 4 rows. Then we merge these single-step explored area by using the spatial transformer and max-pooling, the same way as when we aggregate the 2D object map in our semantic representation. We finally mark the center of the aggregated explored area as 2, indicating where the agent is standing at. An illustration is shown in Figure 6 (for simplicity, we only draw a $3 \times 2$ binary mask), where four single-step explored area maps are aggregated into one. The CNN used to process the explored area is of 4 layers (filter size 3, stride size 2, number of features as $128, 128, 64, 32$). The output of the CNN is flattened and fed into a 2-layer FC network (number of features are $128, 32$). Then we concatenate its output with the output from the multimodal transformer (which extract features for the other 3 modalities) and feed it into the final 3-layer FC network to predict the next exploration action (number of features are $256, 128, N_{exp} + 1$ where $N_{exp} = 3$ and the extra 1 is for the `Stop` action).

**Training Data**   We regenerate the trajectories from the training data of ALFRED by ignoring all object interaction actions and `LookUp/Down`. We train the exploration module by imitation learning on this new training set. Each sample in the new training set corresponds to one navigation subgoal in a trajectory of the original training set. As each trajectory in the training data in AL-FRED starts with an initial horizon of $30°$, all visual observations used for training are of such horizon (i.e., vertical camera angle).

**Hyper-parameters**   We train the exploration module by Adam optimizer with an initial learning rate of $0.001$ and a linear decaying schedule (kicking in only for the second half of the training) to $0$ for a total of 20 epochs. We find the best checkpoint using the coverage and coverage efficiency computed on the valid seen and valid unseen data of ALFRED.

### C.2   ACTION AUGMENTATION

**Zigzagging**   As introduced in the paper, we inject two `LookUp` or two `LookDown` actions alternately after every exploration action so that we acquire images of 3 different horizons at each $(x, y, r)$ observed during exploration. We choose this zigzagging scheme as it is the most efficient way to perform exploration in the vertical camera angle space. An example is listed below with the action sequence (after the periodic injection of `RotateRight`) shown in the first line and the zigzagged sequence shown in the second line.

- Move, Right, Move, Right, Right, Right, Right, Move, Left
- **Down, Up, Up**, Move, **Down, Down**, Right, **Up, Up**, Move, **Down, Down**, Right, **Up, Up**, Right, **Down, Down**, Right, **Up, Up**, Right, **Down, Down**, Move, **Up, Up**, Left, **Down, Down**

The injected actions are bolded, with the first `LookDown` inserted to handle the beginning of the exploration.

**Other Details**    Since each augmented trajectory (i.e., the sequence injected with `RotateRight` and `LookUp/Down`) is a strict superset of the original unaugmented ones predicted directly via the multimodal exploration module, these injection does not interfere with the normal inference pipeline of the exploration module. Specifically, we mask out the inputs (observations, actions history, and the explored area) corresponding to the injected actions in all of the 4 branches.

# D    Other Modules or Models

## D.1    Subgoal Parser and Target Object Parser

**Network Architecture**    We use a transformer-based architecture similar to the multimodal transformer used for exploration. Since the input to both the subgoal parser and the target object parser is the language instruction, we mask out the inputs to the other 3 branches (i.e., we use uni-modal transformers). The two models share the same architecture while not sharing the weights. The subgoal parser is essentially a binary classifier. The target object parser, on the other hand, predicts two pieces of information, namely the target object for the navigation subgoal and its container (if it has one). This prediction involves two steps; we, therefore, model it in an auto-regressive manner.

**Model Training**    We train the subgoal parser and the target object parser by minimizing the cross-entropy loss using the ground truth object information. Specifically, we use APIs provided by AI2Thor to acquire spatial relationships and use it to decide the container object type for each instance of the small objects (which the target object parser is required to predict). We train the 2 models by Adam optimizer with an initial learning rate of $0.001$ and a linear decaying schedule (kicking in in t=the second half of the training) to $0$ for a total of $10$ epochs. We find the best checkpoint using the prediction accuracy (both for the binary prediction problem and the 2-step classification problem) evaluated on the valid seen and valid unseen data of ALFRED.

**Pre-trained Mask RCNN**    We use a pre-trained Mask RCNN in multiple occurrences in our framework. We directly use the checkpoint provided by E.T., which is trained for predicting the instance and segmentation masks of objects in ALFRED.

## D.2    Online Planner

During online planning, we always keep track of the agent's current position, pose, and so on (to be more precise, the $(x, y, r, h)$ tuple). Then at each navigation subgoal, we can perform path planning using Dijkstra's algorithm given the acquired waypoints in our affordance-aware semantic representation. To decide the target horizon of the agent, we perform online exploration to cover 6 different vertical camera angles and select the one with the largest mask area for the target object type (predicted by a pre-trained model, with confidence $> 0.8$). We also add a backtracking mechanism in case that the estimated best horizon $h^*$ does not work for the subsequent object interaction subgoals. In specific, if the subsequent object interaction subgoal fails (i.e., the agent encounters an action failure), we find another horizon and perform inference for the object interaction subgoal again. In total, we try 3 times for $h = h^*, h^* + 15°, h^* - 15°$.

# E    Ablation Studies

**Details for Ablation Studies I**    For all the four variants, we stop the exploration process if (1) the module predicts a `Stop` action, or (2) the number of action failures reaches 4, or (3) the total number of exploration steps (including the injected actions) reaches 500. In the variant AMSLAM + rand.

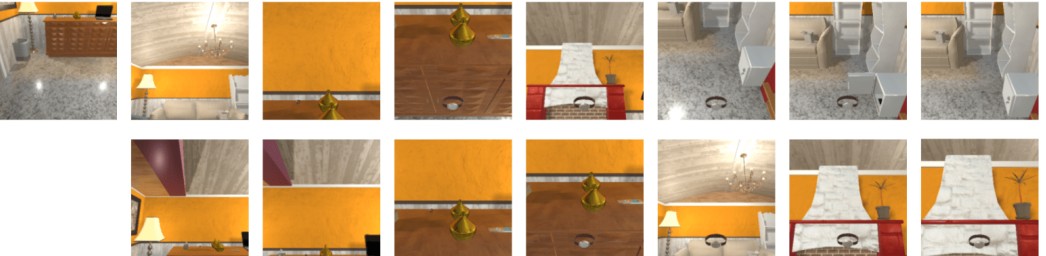

Figure 5: For the task "Move a watch to the inside of a small safe", the first row and second row corresponds to trajectories predicted by ours and E.T. Each column corresponds to a different time step with $t = 0, 6, 12, 18, 24, 32, 35, 36$ from left to right.

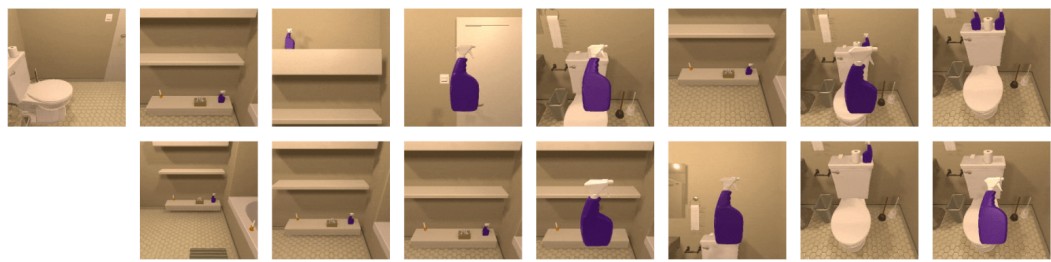

Figure 6: For the task "Place two spray bottles on a toilet tank.", the first row and second row corresponds to trajectories predicted by ours and E.T. Each column corresponds to a different time step with $t = 0, 5, 10, 15, 20, 30, 35, 41$ from left to right.

exploration, we adopt the following random exploration strategy. First, we make 4 `RotateRight` and adjust the horizon of the agent to acquire a $360°$ and 3 horizon view of the environment. Next, we obtain the navigable area estimated by our Neural SLAM system. Then, we randomly sample a point on the boundary of navigable area and navigate to that point (during which process we inject the `RotateRight` and `LookUp/Down` similar to our multimodal exploration strategy). We repeat the previous step until the aforementioned stop condition is satisfied.

**Details for Ablation Studies II** For the variant AMSLAM + rand. horizon, we first disable the online exploration and backtracking strategy of the planner for finding the best horizon $h$. We then randomly choose a final horizon for each navigation subgoal as one of $\{60°, 45°, ..., -15°, -30°\}$.

**Remark**: We will release the code as well as relevant model files soon.

# F QUALITATIVE EVALUATIONS

We also illustrate qualitative results from our proposed method compared to the E.T. baseline. We display a pair of trajectories predicted from both models (for ours, we only show results from the execution phase) for data in the valid unseen and test unseen split, respectively. By utilizing task-driven exploration to acquire the affordance-aware map of the scene, together with the planning and object interaction modules, our model is capable of completing long-horizon instruction-following tasks. In the first task where it asks the agent to grab a watch and then find and store it inside a safe, E.T. fails to navigate to a position such that it can successfully open the safe, whereas ours succeeds. In the second task for moving two spray bottles from a single shelf to the same toilet tank, E.T. fails to find the second sprayer while ours can move both sprayers to the right place.

