# OpenReview forum: "Learning to Act with Affordance-Aware Multimodal Neural SLAM"
_ICLR.cc/2022/Conference — ICLR 2022 Submitted_

### Official Review · Reviewer_kMKr · 2021-10-21

**Correctness:** 4
**Technical Novelty And Significance:** 2
**Empirical Novelty And Significance:** 3
**Recommendation:** 5
**Confidence:** 4

**Main Review:**

**Strengths**

- The method achieves the state-of-the-art performance on the ALFRED benchmark, which is quite challenging.

- The set of ablation experiments are informative.


**Weaknesses**

- The paper is heavily engineered for the ALFRED benchmark. I am not sure if any of these modules can be used for generic embodied tasks.

- The paper makes a big deal about the "affordance-aware" representation in the introduction and other places in the text, but it is actually a simple heuristic that the agent backs up a few steps when it is next to an object. According to the text "the agent will back up 3 steps if the target object is a fridge, 2 steps if it is a safe, a cabinet or a drawer, and 1 step for everything else". These types of heuristics are not generalizable or scalable. The affordance-aware representation should be learned instead.

- There is a strong assumption that "we know the changes (∆x, ∆y, ∆r) of the agent after each action". These types of assumptions are valid only in simulation. Several previous works have made similar assumptions. However, those approaches are suitable only for simulation. In reality, the estimates for agent pose are quite noisy. I would like to see how robust the proposed approach is against noise in the agent pose.

- It is not clear if the exploration steps and the steps used for capturing panoramic views are included in the evaluations.

**Summary Of The Paper:**

The paper proposes a method for solving the ALFRED task (following language instructions to perform a set of household tasks). The model has two main components: (1) Affordance-aware Semantic Representation (2) Multi-modal exploration. The former component estimates the location of the objects in the scene. The latter provides an exploration strategy by using instructions, images, previous actions and previously explored areas as input.

**Summary Of The Review:**

The paper proposes an approach which is heavily engineered for the ALFRED benchmark and I do not see a major novelty that generalizes to other embodied tasks. The method is mainly a mixture of heuristics and a set of pre-trained components that are glued together for the ALFRED tasks. However, it achieves good performance on the ALFRED benchmark, hence, my borderline rating.

---

> ### Author Response · Authors · 2021-11-20
> **Author Response to Reviewer kMKr**
>
> ### 1. "The paper is heavily engineered for the ALFRED benchmark. I am not sure if any of these modules can be used for generic embodied tasks."
>
> We follow the common practices in the literature. Most previous SOTA methods  (LWIT, E.T., HLSM) on ALFRED only evaluate their performance on the ALFRED challenge. Moreover, we believe the idea behind our multimodal exploration design and waypoint-based semantic representation is applicable to generic embodied tasks.
>
> ### 2. "Is the 'affordance-aware' representation actually a simple heuristic that the agent backs up a few steps when it is next to an object? These types of heuristics might not be generalizable or scalable."
>
> The backing up heuristics refers to a simple yet effective design in estimating the waypoints for large objects.
> Waypoints (described at the beginning of Sec. 4) of objects are the key component in our affordance-aware representation.
> We initially tried to directly learn these waypoints for all objects yet we faced a significant challenge.
> We then propose to handle large and small objects differently (perception of small objects is a general challenge in computer vision).
> For large objects (fridges, cabinets, etc.), such simple heuristics are effective and general enough once the location of the object and the navigable area is estimated.
> For small objects, we adopt a general search procedure that is category-agnostic.
> The details are in Sec. 4.2.
>
> ### 3. "There is a strong assumption that 'we know the displacement changes of the agent after each action'. Several previous works have made similar assumptions. However, those approaches are suitable only for simulation. In reality, how robust the proposed approach is against noise in the agent pose?"
>
> In the presence of motion noise and pose sensor noise, Active Neural SLAM shows that these can be solved to an acceptable level by simply adding a pose estimator.
>
> ### 4. "It is not clear if the exploration steps and the steps used for capturing panoramic views are included in the evaluations."
>
> Yes, they are all included. Otherwise, the agent could simply perform an exhaustive search of the environment with no limit on the total number of steps.

---

> > ### Comment · Reviewer_kMKr · 2021-11-28
> > **Lowering the rating**
> >
> > I did not find the response helpful:
> >
> > - I did not ask for evaluating on another benchmark. My point was that the design choices are tailored specifically for ALFRED and do not generalize to other embodied AI tasks.
> >
> > - Heuristics such as "backing up a few steps" should be replaced by learning-based approaches. Also, referring to them as "affordance-aware representation" is just exaggeration.

---

### Official Review · Reviewer_pomV · 2021-11-01

**Correctness:** 3
**Technical Novelty And Significance:** 2
**Empirical Novelty And Significance:** 3
**Recommendation:** 5
**Confidence:** 3

**Main Review:**

strengths:
- results are much better than state-of-the-art and the improved margins are significant.
- the proposed technical module of learning an affordance-aware semantic representation is valid, reasonable, and somewhat novel.
- the system designs are really solid in mixing up many well-designed heuristics and learned modules into a big system that produces very good results.

weaknesses:
- my biggest concern is that the paper presentation is not selling the framework well as an easily-readable research paper, for example, a) there are no qualitative figures showing the performance of the work, only tables with numbers are presented. It is quite important to show qualitative results and your qualitative comparisons to baselines, additionally with some analysis figures clearly proving the effectiveness of each proposed module, for readers to be better convinced where your superior performance comes from. b) the writing for Sec. 4 can be improved. The current writing is quite messy mixing up the two contributions together. Also, many important details (e.g. the architecture to learn the affordance maps and the aggregation steps) are put in supplementary. c) for the experiments, the authors spent many more words on Sec 5.1 and 5.2 than Sec 5.3 (the main results). The main results subsection is very short and does not contain analysis over the tables or show qualitative figures to analyze the results.
- another of my major concerns is whether or not using the language instructions during the exploration stage is a reasonable and realistic setting. My opinion is that we should not assume we know the task (goal) and the instructions for subgoals beforehand. and during the exploration. Otherwise, we need different exploration steps given different tasks. Or, at least, you need to report your exploration steps as part of the task execution steps, since you need different exploration steps given any new test task. Please clarify if I'm wrong with this. But, the confusion regarding this point prevents me from recognizing the claimed technical contribution on leveraging multimodal inputs during exploration.
- though there are some novel designs for the affordance learning part, there are previous works (Qi et al, 2019, and Nagarajan & Grauman, 2020) that have proposed the essential ideas, and I don't see too much difference other than adding some heuristics-derived waypoint definitions. Please clarify if I missed or misunderstood the main differences from previous works.
- though I recognize that the presented system achieves amazing results and significantly refreshes the state-of-the-art, it's unclear how much performance gains come from combining many state-of-the-art architectures, say transformers or bert, and how much comes from the claimed two novel technical design points. Can you compare to baseline methods with the same architectures or report the parameter amounts for different models, to give us a sense of this?
- one minor issue is to use \citep instead of \cite in latex for a more valid citation format of papers

**Summary Of The Paper:**

This paper proposes a new framework that improves the state-of-the-art performance of the ALFRED benchmark (accomplishing navigation and interaction tasks given language instructions in AI2THOR environments) by 40% relatively. Two claimed main technical contributions are: 1) leveraging multimodal inputs, specifically the newly incorporated language instructions and visited area maps, during exploration, 2) proposing an affordance-aware semantic representation that marks the object locations, heights, and possible agent interaction spots. Results that are 40% relatively better than the state-of-the-art are observed and several ablation studies over key network modules are also presented.

**Summary Of The Review:**

Overall, I have a mixed feeling. On one hand, the performance is really good and significant. On the other hand, the two claimed technical contributions are not stated clearly about their significance of differences than the previous methods or the validity of the design. Also, this paper is more like a system report other than a well-written research paper with clearly presented and analyzed novel techniques. Adding qualitative figures and analysis will definitely help on this front.

---

> ### Author Response · Authors · 2021-11-20
> **Author Response to Reviewer pomV**
>
> ### 1. There are no qualitative figures and comparisons showing where the superior performance of the work comes from.
>
> Thanks for the constructive suggestions, we will add them in the updated version.
>
> ### 2. "The writing for Sec. 4 can be improved. The current writing is quite messy mixing up the two contributions together. Also, many important details are put into the supplementary.
>
> We will reorganize section 4 for a better presentation.
>
> ### 3. "We should not assume we know the task (goal) and the instructions for subgoals beforehand during the exploration. Otherwise, we need different exploration steps given different tasks. Please clarify. The confusion regarding this point prevents me from recognizing the claimed technical contribution on leveraging multimodal inputs during exploration."
>
> We would like to clarify this. We do have different exploration steps for different tasks and so the language instructions can indeed help to make exploration more efficient. In other words, the agent always performs task-driven exploration. Also, the exploration steps count towards the total task steps. We set a hard limit of 700 steps for the exploration phase (notice that the total number of steps including both phases have a limit of 1000).
>
> ### 4. "There are previous works (Qi et al, 2019, and Nagarajan & Grauman, 2020) that have proposed the essential ideas about affordance-awareness, and I don't see too much difference other than adding some heuristics-derived waypoint definitions."
>
> The approach in Qi et al works with RGB-D data assuming direct access to the ground truth depth information, which is significantly easier than the setup in ALFRED where the agent does not have such access.
> To tackle the inaccuracy for predicting affordance information of the small objects, we design to handle large and small objects differently (described in Sec. 4.2).
> More importantly, their method acquires the affordance model as a segmentation module which does not directly give you the position and orientation (i.e., waypoints) for the agent to interact with the objects.
> For tasks involving long-horizon planning of both navigation and interactions, these waypoints can be very helpful.
>
> On the other hand, the method in Nagarajan & Grauman, 2020 only focuses on pure navigation problems. Whereas, in ALFRED, the presence of subsequent object interactions requires a different kind of affordance map than in Nagarajan & Grauman, 2020 and poses a larger challenge for affordance-aware map acquisition.
>
> ### 5. "Though the presented system achieves amazing results and significantly refreshes the state-of-the-art, it's unclear how much performance gains come from combining many SOTA architectures or from the claimed two novel technical design points."
>
> The Transformer structure used in multiple parts of our system is of the same scale as the one used in HiTUT and E.T.
> Specifically, all the transformer architectures we use are modified from E.T. (each with roughly the same or less total number of parameters).
> As a result, we consider E.T. to be a reasonable baseline model to highlight the advantages of our system.
> We will add full implementation details to the Appendix.

---

> > ### Comment · Reviewer_pomV · 2021-11-29
> > **Keep my rating**
> >
> > I first want to thank the authors for the replies. I have also read the other reviews and the paper revision.
> >
> > The authors have addressed some of my concerns:
> >   -  the comparison fairness regarding the backbones;
> >   -  adding qualitative results and analysis;
> >   -  clarifying that the exploration steps are included in the final evaluation steps.
> >
> > However, I remain my borderline reject rating and also very much agree with the other reviewers, since
> >   -  this paper has many heuristic designs that are over-tuned to the ALFRED benchmark. The designs are very local so it's hard to imagine it will work to other benchmarks/real-world scenarios.
> >   -  the major technical contributions are not clear. There are so many small designs. Although I understand that they are important for improving the numbers practically, it does not help make this paper a good research work.
> >   -  the paper writing needs major revisions.

---

### Official Review · Reviewer_2ghY · 2021-11-04

**Correctness:** 2
**Technical Novelty And Significance:** 2
**Empirical Novelty And Significance:** 2
**Recommendation:** 3
**Confidence:** 4

**Main Review:**

Strengths:
- This submission is well motivated. Training robotic assistants to follow human instructions to complete an interactive task is an important and challenging problem.

- The proposed method achieves good performance, improving 40% over prior published approaches to achieve 23.48% success rate.

Weaknesses:
- The writing clarity and readability can be improved significantly. The method seems extremely complicated and is very difficult to understand. There are way too many forward and backward references in Section 4. Even after reading over the methods section twice and going over the details in the Appendix, I am unable to understand many details in the method. For example:
    - In sec 4, what is meaning of module operating at the "subgoal level"? In contrast to what in the E.T. paper? Why is the explored area considered an extra modality?
    - In sec 4.1, how does the agent predict the task described in the language instructions? How does the agent know the subgoals in each task? How does the agent switch between exploration and execution phases? What is the difference between planned and prediction actions?
    - In Figure 1, it is unclear where and how the semantic map is used. It is unclear where the explored area is coming from (my understanding is that the environment does not provide the explored area). The caption refers to "affordance-aware semantic representation" which is not shown in the figure. What is the meaning of the "actual" task?
    - In Sec 4.2, what are the delta rule-based functions?
    - In Sec 4.3, how does the agent know the which low-level instruction corresponds to the current navigation and subsequent object interaction subgoals?

- The method seems to use of lot of arbitrary choices, rules and hacks. For example, "we define a single-step explored region as a binary map where a 5×3 rectangle grid (as a hard-coded field of view)", "We manually inject 4 RotateRight after every 2 MoveAhead predicted by our module to acquire 360◦ views of the scenes", "we further inject two LookUp or two LookDown actions alternately after every exploration action", "The agent then goes through 6 horizons {60◦, 45◦, ..., 0◦, −15◦} and obtains the mask prediction (with confidence > 0.8)". All these choices seem to be specific to the Alfred benchmark. I can not imagine a robot operating in the real-world in this manner. My worry is that the authors have exploited unrealistic approximations in simulation environments (such as discrete action and state space, no noise in motion and pose sensors and use of high-level interaction actions), and over-optimized the design choices specific to this benchmark, which are likely to not result in better performance in more realistic settings.

- The technical contributions of this paper are unclear:
   - The authors claim in the introduction "we propose the first multimodal exploration module that takes language instruction as guidance and keeps track of visited regions." I believe the Active Neural SLAM model, which AMSLAM is based on, also keeps track of visited regions to learn exploration. Based on authors' definition, even Active Neural SLAM is multimodal as it used pose, explored area and visual observations as input. Is the addition of language instructions as input the only change to the model?
    - The authors claim they introduce "Affordance-aware semantic representation that estimates object positions, heights, and relative spatial relationships". I believe the semantic representations in Chaplot et al. 2020b and Blukis et al. 2021 also estimate object positions, heights and relative spatial relationships. It is unclear what makes the proposed representation "affordance-aware".

**Summary Of The Paper:**

This paper presents a Neural SLAM-based approach for tackling embodied multimodal tasks in ALFRED benchmark. The approach, called Affordance-aware Multimodal Neural SLAM (AMSLAM), utilizes several modalities for exploration, predicts an affordance-aware semantic map, and plans over it at the same time. The approach achieves 40% improvement over prior published work.

**Summary Of The Review:**

Overall, the motivation is good and the method achieves a good performance on a public ALFRED benchmark. However, the authors seem to have made many design choices specific to this benchmark which are likely to not result in better results on realistic tasks. The technical contributions of this paper are unclear and the writing clarity needs to be improved significantly.

---

> ### Author Response · Authors · 2021-11-20
> **Author Response to Reviewer 2ghY (part I)**
>
> Despite recognizing the good motivation and great performance achieved by our method on the ALFRED challenge, the reviewer has three major concerns. Let us respond one by one.
>
> # Major Concern I: "The method seems to use a lot of arbitrary choices, rules and hacks specific to the Alfred benchmark"
>
> Some of the choices (e.g., horizon angles) are regulated by the AI2Thor simulator and the ALFRED challenge, which are adopted commonly by a significant body of published related work on building models for this benchmark.
> We present our choices in great detail to facilitate reproducibility and our design choices are backed by comprehensive experiments.
> We do agree that the presentation could be improved and we are updating the paper for flow and readability accordingly.
> Please find our detailed responses as follows:
>
> ### 1. "define a single-step explored region ... where a 5×3 rectangle grid ..."
>
> We find that the exact shape of the single-step explored region does not matter much (we have tried 3x2, 4x2, 5x4).
> As long as it helps to track where the agent has visited, this extra modality improves the exploration module as demonstrated in Sec. 5.4.
>
> ### 2. "manually inject 4 RotateRight after every 2 MoveAhead predicted by the module...", "further inject 2 LookUp or 2 LookDown..."
>
> Injecting RotateRight periodically to acquire a panoramic view of the indoor environment is not an uncommon practice (e.g., also found in previous methods such as LWIT and HLSM).
> Injecting LookDown or LookUp actions is also not an arbitrary choice.
> It is a trade-off between exploration and exploitation.
> On one hand, it provides a better view of the environment.
> On the other hand, it does not significantly increase the number of actions in the exploration phase (which can not exceed the upper limit of 1000 together with all actions in the execution phase).
>
> ### 3. "the agent then goes through 6 horizons and obtains the mask prediction (with confidence > 0.8)"
>
> Since the action space is discrete in AI2THOR, we choose to perform a search over most horizons allowed for the agent.
> We choose the confidence threshold 0.8 by selecting from {0.5, 0.6, 0.7, 0.8, 0.9} on the valid_unseen data.
> We will add these missing details to the Appendix.
>
> ### 4. "The authors have exploited unrealistic approximations in simulation environments (such as discrete space, no noise in motion and high-level interaction actions)."
>
> Our experiments focus on the ALFRED dataset, which simplifies complex object manipulation and locomotion into high-level actions and discrete state space.
> Low-level object manipulations are not the focus of our paper as there are other benchmarks specifically designed for such tasks, e.g., ManiSkill (https://arxiv.org/abs/2107.14483).
> In the presence of motion noise, Active Neural SLAM shows that simply adding a pose estimator can tackle this acceptably well.
>
> # Major Concern II: "The technical contributions of this paper are unclear"
>
> ### 1. "Active Neural SLAM also keeps track of visited regions to learn exploration. Is the addition of language instructions as input the only change to the exploration model?"
>
> There are 2 major differences between Active Neural SLAM and the SLAM module in our method.
> (1) Active Neural SLAM focuses on pure navigation problems. The exploration module in our method tackles map acquisition for both navigation and manipulation, which is significantly harder and requires affordance information.
> (2) Active Neural SLAM focuses on maximizing the coverage of the environment with no semantic map acquisition and is trained via RL. Ours are much more sample efficient with a focus on task-driven exploration, where the actions are guided by language instructions and the result is a semantic map with affordance information.
>
> ### 2. "The representation used in Chaplot et al. 2020b and HLSM also involves object positions, heights, and so on. What makes the proposed method different so that it claims to be affordance-aware?"
>
> Our major difference is that we devise a waypoint-oriented representation instead of trying to estimate the exact 3D coordinates of each object in the scene. Each object in our representation is constructed by the associated location from where the agent can interact with it (illustrated in Figure 2). This is easier to obtain and better suited for the downstream object interaction tasks than directly estimating the locations of all objects. To acquire such a map we handle large and small objects differently as described in Sec. 4.2.

---

> > ### Author Response · Authors · 2021-11-20
> > **Author Response to Reviewer 2ghY (part II)**
> >
> > # Major Concern III: "Readability needs to be improved significantly. The method seems extremely complicated."
> >
> > To improve readability we will reorganize the writing in section 4 and make several other changes throughout the paper.
> > We believe that compared to previous SOTA methods (e.g., HLSM) on ALFRED, our method is actually simpler.
> > Please find our point-to-point response as follows:
> >
> > ### 1. "In sec 4, what is the meaning of module operating at the subgoal level compared to E.T.?"
> >
> > In contrast to E.T. which takes inputs for the entire task all at once and predicts the actions accordingly, our approach operates (like HLSM) by taking inputs for each subgoal and predicting the associated actions one by one.
> >
> > ### 2. "Why is the explored area considered an extra modality?"
> >
> > The “explored area” used as input to the Transformer module for the exploration phase helps to keep track of the area visited by the agent in a geometric and explicit way.
> > The ablation studies (see Sec. 5.4 & Tab. 4) empirically show the advantage of this extra modality.
> >
> > ### 3. "In sec 4.1, how does the agent predict the task described in the language instructions? How does it know the subgoals and which low-level instruction corresponds to the current subgoal?"
> >
> > As introduced in Sec. 3, the input format of tasks in ALFRED consists of a sequence of low-level instructions, each corresponding to a subgoal.
> > In Sec. 4.1, we introduce that the agent processes each instruction to predict whether each subgoal is for navigation or not and the target object types for the navigation subgoals, by using the subgoal parser and the target object parser. The details of these modules are in section 4.4.
> > See Fig. 1 (right) as a high-level illustration of the data flow in the execution phase.
> > We will reorganize Sec. 4 to improve the presentation.
> >
> > ### 4. "How does the agent switch between exploration and execution phases?"
> >
> > As described in Sec. 4.4, we always perform exploration first and then execution (w\o back-and-forth).
> > In the exploration phase, the agent predicts exploration actions a ∈ {MoveAhead, RotateLeft, RotateRight, Stop} auto-regressively for each navigation subgoal.
> > Every time a Stop action is predicted, the agent switches to the next subgoal.
> > The exploration phase ends after the last navigation subgoal is finished.
> >
> > ### 5. "What is the difference between planned and prediction actions?"
> >
> > In the execution phase, we refer to the actions obtained from the planner as “planned actions” for the navigation subgoals and actions obtained from the Object Interaction Transformer as “predicted actions” for the non-navigation subgoals.
> >
> > ### 6. "In Fig. 1, it is unclear where and how the semantic map is used."
> >
> > As displayed in Fig. 1 (right), in the execution phase, the semantic map acquired from the exploration phase is used in the planner to generate actions for the navigation subgoals.
> >
> > ### 7. "In the exploration phase, it is unclear where the explored area is coming from (my understanding is that the environment does not provide the explored area)."
> >
> > The “explored area” comes from the agent’s prediction of its exploration actions.
> > Given such exploration actions, we can directly compute where the agent has visited.
> >
> > ### 8. "The caption of Fig. 1 refers to "affordance-aware semantic representation" which is not shown in the figure."
> >
> > The "affordance-aware semantic representation" is illustrated as the “semantic map” in the exploration phase and used by the planner in the execution phase.
> > It is illustrated in Fig. 2 with more details.
> > We will adjust the wording and the diagram to make this clearer.
> >
> > ### 9. "What is the meaning of the 'actual' task in the caption of Fig. 1?"
> >
> > We refer to the sub-goal level tasks executed in the execution phase, e.g., pick up the book, in contrast to the exploration performed in the exploration phase.
> >
> > ### 10. "In Sec 4.2, what are the delta rule-based functions?"
> >
> > These are the backup functions described in Appendix A.2. We will add full implementation details.

---

### Author Response · Authors · 2021-11-23
**The revised version has been uploaded**

We have uploaded the revised version with modifications throughout the paper and the appendix (marked red). We add more details, incorporate necessary clarifications and figures for qualitative results into this version.

---

### Decision · Program_Chairs · 2022-01-20

**Decision:**

Reject

**Comment:**

This paper presents a SLAM based approach for the ALFRED benchmark. The presented method, Affordance aware Multimodal Neural SLAM has two key advantages over past works: It uses a multimodal exploration strategy and it predicts an affordance aware semantic map. It also obtains a very large performance improvement over the ALFRED benchmark. The reviewers for this paper were quite impressed by the large improvements obtained by this technique. However, there were two major concerns across the reviews: (1) Are the design choices made in this paper heavily engineered towards ALFRED ? (2) Does the work make too many assumptions about the setting (unrealistic assumptions that may not really hold in more realistic environments or the real world) ? The authors have provided a detailed response and answered many questions posed to them, but the reviewers continue to have concerns about the generalizability of the proposed method. Another point of concern pointed out by a reviewer is whether it is reasonable in a realistic setting to perform exploration with a knowledge of the downstream task. This point has not really been answered satisfactorily by the authors. My takeaway is that the method presented by the authors clearly works on ALFRED. But it contains several design choices that are largely ALFRED specific and in some cases unrealistic. This provides fewer benefits to readers looking for more general insights that can be valuable across a suite of tasks. As a result of this, and in spite of the large gains, I recommend rejecting this paper.